# The Roles of Septins in Regulating Fission Yeast Cytokinesis

**DOI:** 10.3390/jof10020115

**Published:** 2024-01-30

**Authors:** Shengnan Zheng, Biyu Zheng, Chuanhai Fu

**Affiliations:** 1MOE Key Laboratory for Cellular Dynamics & Center for Advanced Interdisciplinary Science and Biomedicine of IHM, Division of Life Sciences and Medicine, University of Science and Technology of China, Hefei 230027, China; biyu@mail.ustc.edu.cn; 2Anhui Key Laboratory of Cellular Dynamics and Chemical Biology & Hefei National Research Center for Interdisciplinary Sciences at the Microscale, School of Life Sciences, University of Science and Technology of China, Hefei 230027, China

**Keywords:** *Schizosaccharomyces pombe*, septin, cytokinesis

## Abstract

Cytokinesis is required to separate two daughter cells at the end of mitosis, and septins play crucial roles in many aspects of cytokinesis. While septins have been intensively studied in many model organisms, including the budding yeast *Saccharomyces cerevisiae*, septins have been relatively less characterized in the fission yeast *Schizosaccharomyces pombe*, which has proven to be an excellent model organism for studying fundamental cell biology. In this review, we summarize the findings of septins made in fission yeasts mainly from four aspects: the domain structure of septins, the localization of septins during the cell cycle, the roles of septins in regulating cytokinesis, and the regulatory proteins of septins.

## 1. Septins in General

Septins are a group of highly conserved GTPases that are capable of associating with the membranes in eukaryotes, but they are not present in higher plants [1,2]. Dysfunction of human septins has been linked to several diseases, including hereditary neuralgic amyotrophy [3], lymphoma [4], breast and ovarian cancers [5], and male infertility [6]. Septins have a similar domain structure across different species, consisting of a central GTP-binding domain, a variable N-terminal region, and a coiled-coil C-terminal region. In general, the N-terminal region is rich in basic residues (i.e., the polybasic region, PBR) and can associate with phosphatidylinositol-4,5-bisphosphate (PIP2) on the membrane [7,8,9]. In addition, septins interact with each other via two different interfaces. (1) The N- and C-termini of two neighboring septins interact with one another to form an NC interface [10,11]; (2) the two GTP-binding domains of two neighboring septins interact with one another to form a GG interface. Therefore, using the NC and GG interfaces, septins can assemble into complexes, which then form filaments on membranes by annealing in an end-to-end manner [12,13].

In the budding yeast *Saccharomyces cerevisiae*, purified septins are organized as linear symmetric hetero-octamers in vitro, which can further form paired “tracks-like” filaments under a low salt condition or in the presence of PI(4,5)P2 [9]. In *Caenorhabditis elegans* and human cells, septins are organized as linear symmetric hetero-tetramers and hetero-hexamers, respectively, both of which can form nonpolar filaments in vitro [14,15]. However, how purified *Schizosaccharomyces pombe* (fission yeast) septins are organized awaits further investigation. 

Interestingly, budding yeast septins also form curved bundle structures, in addition to symmetric rod structures described above, by using septin subunit Shs1 [16]. This finding suggests that specific septin subunits could be tailored to build characteristic septin structures. Moreover, specific regions/domains have been found within some septin subunits to mediate distinct functions of septins. For example, amphipathic helices are present in some types of septins and enable septin structures to recognize and sense positive micrometric high curvatures of membranes on the cell cortex, including the cleavage furrow formed during cytokinesis, the base of branches, and the edge of protrusions [17,18]. On these high curvature membranes, septin filaments can be further organized into higher-order structures, such as bars, rings, or hourglass-like structures [19]. These higher-order septin structures have multiple functions, including serving as diffusion barriers to compartmentalize membranes [20,21], increasing the rigidity of the plasma membrane to alter cell morphology [22,23], and serving as scaffolds to bring together various cytoplasmic and cytoskeletal proteins. A notable example of the scaffold function of septins is seen in the mother–daughter neck of budding yeast, where most proteins localize in a septin-dependent manner [24,25,26]. However, septins also form filaments or rod structures on low curvature membranes, such as the fission yeast lateral cortex [27,28], and the hyphal cortex in the filamentous fungus *Ashbya gossypii* [17]. Why septins form different structures on membranes with different curvatures and how different septin structures dynamically remodel still need to be addressed in future studies.

Septins were first discovered in the budding yeast *Saccharomyces cerevisiae* and were found to play a crucial role in cell division [29]. While the organization and function of septins in budding yeast have been extensively studied [19,30,31], the organization and function of septins in the fission yeast *Schizosaccharomyces pombe*, which has proven to be an excellent model organism for studying the cell cycle [32], the cytoskeleton [33], cell polarity [34], and organelle biology [35,36], were relatively less characterized. Fission yeast is a rod-shaped unicellular organism [32], and the rod-shaped morphology makes fission yeast possess two different membrane curvatures: (1) the low curvature at the lateral sides of cells, and (2) the high curvature at the growing tips and the cleavage furrow. These morphological features enable the analysis of septin behaviors on membranes with different curvatures in one organism. In addition, fission yeast grows by tip elongation [32], and the length of the cell reflects the phase of the cell cycle. Thus, fission yeast is an excellent model to study septins throughout the cell cycle. Moreover, the highly polarized growth pattern of fission yeast makes it useful for studying the roles of septins in regulating cell polarity. Hence, fission yeast is an excellent model organism, complementing budding yeast for studying septin biology.

Similar to most eukaryotes, the assembly and constriction of the contractile actomyosin ring (CAR) is a key process of cytokinesis in fission yeast. During CAR constriction, a septum structure comprising α- and β-glucans is synthesized in the middle of the cell with a primary septum flanked by two secondary septa [37,38]. Subsequently, as the primary septum is degraded by glucanases, the two daughter cells separate completely [39,40]. For further details on cytokinesis in fission yeast, please refer to published review articles [41,42,43]. Although septins are not essential in fission yeast, the improper assembly of septin rings leads to a defect in septation [44,45,46,47], suggesting that septin rings play crucial roles in regulating cytokinesis in fission yeast. In this review, we focus on fission yeast septins and discuss their roles in regulating cytokinesis.

## 2. Septins in Fission Yeast

Fission yeast contains seven septins, namely Spn1–7. Spn1–4 are expressed during vegetative growth while Spn2 and Spn5–7 are expressed during sporulation [48]. Similar to septins in other eukaryotes, Spn1–7 have a central GTP-binding domain, a variable N-terminus, and a coiled-coil C-terminus (except Spn2 and Spn7) (Figure 1). The N-termini of Spn1, 2, 3, 4, and 7 contain a polybasic region (Figure 1), which is likely responsible for interacting with the membranes. Lipid-binding assays have revealed that Spn2 and Spn7 have an affinity for phosphoinositides, particularly phosphotidylinositol-4-phosphate and phosphotidylinositol-5-phosphate, and phosphotidylinositol-4-phosphate is enriched on the forespore membrane [48]. Furthermore, Spn2, 5, 6, and 7 form a complex in vitro, and mutations of these septins caused the disoriented extension of forespore membranes [48]. These results demonstrate that the septin complex, comprising Spn2, 5, 6, and 7, plays a crucial role in mediating the proper formation of the forespore membrane during fission yeast sporulation.

During vegetative growth, fission yeast cells form a distinct septin complex, comprising Spn1, 2, 3, and 4, to regulate cytokinesis [27]. Affinity purification experiments have revealed that the septin components in the complex form a linear order, i.e., Spn3-Spn4-Spn1-Spn2, with Spn4 and Spn1 serving as the core components. Consistently, the absence of Spn1 or Spn4, but not Spn2 and Spn3, abolishes the cortical localization of the other septins [27,47]. Although Spn2 and Spn3 do not appear to serve as core components of septin complexes, these two components play a role in maintaining the proper assembly of septins. In the absence of Spn2, other septins localize as filaments in the cytoplasm but still form a compact ring at the division site [27,47]. In cells lacking Spn3, the cortical septins disperse and form small puncta on the cortex and in the cytoplasm. In addition, in cells lacking Spn3, the septin ring still forms but the cells show mild cytokinetic defects [27,47]. Interestingly, Spn1 and Spn4 form dot-like structures, but not a compact ring, at the equatorial region of cells lacking both Spn2 and Spn3, and the double-deletion cells show a severe cytokinesis defect, which is similar to the one caused by the absence of Spn1 or Spn4 [27]. These results indicate that although Spn1 and Spn4 serve as core components in the assembly of septin complexes, Spn2 and Spn3 may play different roles in fine-tuning the assembly of septin complexes.

To visualize septins, different septin components were tagged with the fluorescence proteins GFP [27], YFP [27], CFP [45,49], or tdTomato [47]. All these fluorescent tags do not appear to affect the formation of the compact ring structure at the division site during cytokinesis. Most of the studies using the fluorescently tagged septins focus on addressing the localization and function of septins during cytokinesis (i.e., the septin ring structure) [27,45,50,51,52]. However, the localization and function of septins on the cortical membrane during interphase has been relatively less characterized. Our previous work showed that tdTomato-tagged Spn1 displays pronounced localization to the cortex of cells during interphase [28,47], and Spn1-tdTomato did not appear to affect cytokinesis. This cortical localization pattern of septins is consistent with the one shown in several previous publications. For example, Spn1-mEGFP was found to form filaments on the cortical membrane in interphase by annealing [12], Spn1-RFP and Spn3-GFP were found to form puncta/filaments on the cortical membrane in interphase cells [53], Spn3-GFP was shown to form puncta on the cortical membrane in interphase cells [54,55,56,57], and Spn2-GFP was shown to form faint puncta on the cortical membrane in interphase cells [27]. By contrast, YFP-tagged Spn4 did not appear to localize to the cortical membrane in interphase cells [27], and CFP-tagged [49,58] or mEGFP-tagged [46] Spn1 did not appear to localize to the cortical membrane. Therefore, in the future, it is still necessary to further test the interphase cortical localization of septin components tagged with different fluorescent tags. Additionally, the functionalities of the fluorescently tagged septins should be further tested. To confirm the cortical localization of septins in interphase cells, one may perform immunofluorescence experiments by using antibodies against specific septins. Alternatively, septins may be tagged with small tags, which can then be visualized with corresponding antibodies via immunofluorescence.

It is likely that the Spn1–4 complex depends on its N-terminal polybasic regions for localizing to the plasma membrane as the Spn1–4 complex exhibits an affinity for liposomes in vitro [28]. Interestingly, the Cdc42 GTPase-activating protein Rga6 is required to promote the efficient localization of the Spn1–4 complex to the cell cortex [28], and once arriving at the cell cortex from the cytosol, the Spn1–4 complex forms high-order filaments by annealing in an end-to-end fashion [12]. Hence, although in fission yeast cells different septin complexes form to mediate growth-state-specific functions, the modes of the interactions between septins and between septins and the plasma membrane are similar. Further investigation may focus on studying how the septin complex is localized to the cortical membrane, how the cortical localization of the septin complex is regulated in space and time, and how the septin complex is involved in regulating cortical functions, including cell polarity.

## 3. Septin Localization during the Cell Cycle

The localization of septins changes in a cell-cycle-dependent manner. In mammalian cells, septin filaments have an intimate relationship with actin and microtubule cytoskeletons, and are referred to as the fourth cytoskeleton within the cell [2]. During interphase, mammalian septins form filament structures in the cytoplasm (Figure 2A). At anaphase, RhoA is activated by the centralspindlin complex at the midzone to promote cytokinesis [59], and at early telophase, septins on the cell cortex are recruited to the cleavage furrow by anillin, which is the master regulator of septins and functions as a scaffold during cytokinesis. During CAR constriction and intercellular bridge elongation, anillin and septins form a collar-like structure (Figure 2A), whose orientation is parallel to the cell division plane and functions to promote CAR constriction and intercellular bridge elongation [60]. Subsequently, upon the completion of intercellular bridge elongation, the collar structures of anillin and septins retract and remodel to form three distinct rings (Figure 2A), which are vertical to the cell division plane. The central anillin ring is the stem body flanked by double septin rings, which have a narrower diameter and mediate the formation of the secondary ingression sites (Figure 2A). One of the ingression sites ultimately evolves into an abscission site, and non-muscle Myosin-II plays a crucial role in generating the abscission site [61]. Additionally, it has been established that the formation of the abscission site depends on septins. Specifically, septin double rings are responsible for recruiting the ESCRT (endosomal sorting complex required for transport) III at the site to execute abscission and for preventing the premature diffusion of other cytokinetic components [60,62,63]. Hence, septins undergo dynamic and structural changes during mammalian cytokinesis.

The localization and organization of septins are similarly dynamic in budding yeast cells. At the early G1 phase, septins form a patch-like structure at the presumptive bud site (Figure 2B). Before the budding of the daughter cell, the septin patch becomes a ring-like structure, depending on the GTPase Cdc42 and its effectors, and after the budding of the daughter cell (S phase), the septin ring expands into an hourglass-like structure at the mother-bud neck (Figure 2B). Fluorescence recovery after photobleaching (FRAP) experiments revealed that the hourglass-like septin structure is more stable than the septin ring and functions as a scaffold to recruit and concentrate cytokinetic proteins at the neck for CAR assembly [31]. Moreover, the orientation of filaments in the septin ring and the hourglass-like structure is vertical to the cell division plane. At the onset of cytokinesis (telophase), the hourglass-like septin structure splits into two distinct rings, sandwiching the CAR and the septum [64]. After complete separation of the mother and daughter cells, the old septin ring on the mother cell is disassembled while new septins accumulate at the presumptive budding site of the daughter cell to prepare for the next round of cytokinesis [30,31] (Figure 2B).

The dynamic nature of septin localization is similar in fission yeast cells. At interphase, septins form puncta structures on the cell cortex (Figure 2C). The puncta may elongate and accumulate to form larger structures by annealing on the cortical membranes adjacent to the growing tips. Rga6, a Cdc42 GAP (GTPase-activating protein), interacts with septins and promotes the cortical localization of septins [28]. At telophase, septins accumulate to the equatorial cortex and subsequently compact into a septin ring. At the late stage of cytokinesis, the septin ring splits into two distinct rings, and the distance between the double rings is approximately 300 nm [47]. Finally, septin rings are disassembled from the division site during cell septation [47] (Figure 2C). In fission yeast cells, the organization and orientation of septin filaments in the ring structures remain unclear. What signaling induces the accumulation of septins at the equatorial region and how septin rings are assembled await further investigation. 

## 4. The Roles of Fission Yeast Septins in the Assembly, Maintenance, and Constriction of the Contractile Actomyosin Ring

Fission yeast cytokinesis are a series of events taking place in order, with the CAR assembled on the equatorial cortex, maintained, and constricted, and during CAR constriction, a septum structure forms and is subsequently degraded to mediate cell septation [41]. During CAR assembly, cortical nodes, comprising the kinase Cdr2, first emerge at the equatorial region of the cell and serve as precursors for further assembling the CAR [41]. CAR maintenance and constriction depend on the activity of the septation initiation network (SIN), which is analogous to the mitotic exit network (MEN) in budding yeast and the HIPPO pathway in metazoans [65]. Additionally, CAR constriction is facilitated by the synthesis of the primary septum, which is mediated by the glucan synthases Bgs1 and Ags1 [66,67].

Septin localization in fission yeast changes in a cell-cycle-dependent manner. Specifically, septins form puncta on the cortical membrane during interphase and accumulate on the equatorial membrane to form non-contractile septin rings when CAR constriction starts. Research from our group and others has established a connection between septins and the CAR [46,47,50]. In addition, it was demonstrated that septins form a ring after the unconventional myosin-II Myp2 joins the CAR, and the localization of Myp2 on the CAR is independent of septins [52]. Previous studies demonstrated that fission yeast septins play a crucial role in cell septation but a minor role in regulating early cytokinetic events [49,68]. However, using live-cell microscopy, we discovered that in cells lacking the core septin component Spn1, the assembly of CAR nodes is accelerated and the SIN kinase Sid2 and its interacting protein Blt1 appear earlier on the equatorial cortex [47]. Similarly, in cells lacking Spn1, Bgs1 and Ags1, which are responsible for the synthesis of linear β-(1,3) glucans and α-(1,3) glucans, respectively, also emerge earlier at the equatorial cortex [47]. These results support a model in which septin puncta on the cell cortex function to prevent the premature assembly of the CAR and the premature arrival of cytokinetic proteins at the equatorial cortex (Figure 3A).

Septins play a crucial role in promoting CAR constriction in addition to regulating CAR assembly. This is evident by our findings that the absence of Spn1 decreases the rate of CAR constriction and prolongs the disassembly process of the CAR [47]. Our work also shows that the septin ring colocalizes with the ring of Sid2, the SIN kinase regulating CAR assembly and constriction, at the equatorial region during CAR constriction and that Sid2 at the equatorial cortex diffuses away prematurely in cells lacking Spn1 [47]. Similarly, in cells lacking Spn1, the glucan synthases Bgs1 and Ags1 delocalize prematurely from the equatorial cortex and accumulate abnormally at the cell ends [47]. The SIN activity and the synthesis of septum are required for CAR maintenance and constriction [65,66,67]. Therefore, the findings support a model in which septin rings restrict the SIN, glucan synthases, and CAR components at the cleavage furrow to promote CAR constriction and septum formation (Figure 3B). Conversely, the SIN activity and CAR constriction are required for the normal accumulation and compaction of the septin ring structure at the cell division site [50]. In the future, it will be worthwhile to investigate how the septin complex orchestrates the timely arrival of the cytokinetic proteins at the equatorial cortex and whether the septin complex functions as a scaffold during cytokinesis.

In budding yeast, the proper and timely organization of septin structures also play crucial roles in regulating cytokinetic events. For example, forming an hourglass-like septin structure is required to promote CAR assembly [64,69]. As described above, the budding yeast hourglass-like septin ring splits into double rings, sandwiching the CAR at the onset of cytokinesis. Although CAR constriction and septin ring splitting occur approximately at the same time, the exact time of these two events is separated. Specifically, the splitting of the septin ring precedes CAR constriction by 4~5 min [70]. The absence of MEN, the counterpart of the SIN in fission yeast, components results in the failure splitting of the septin ring and stable CAR at the bud neck [70]. Therefore, the MEN and SIN are similarly required to mediate the proper functions of septins and the CAR in budding and fission yeasts. Additionally, the septin hourglass-like structure hampers CAR constriction, and septin ring splitting or the clearance of the septin ring from the division site is required to constrict the CAR [70]. Therefore, the proper organization of septin structures is required to mediate the proper CAR constriction in both budding and fission yeasts.

In budding yeast, the role of septin double rings in regulating septum formation is still unclear. It is possible that the CAR and septin double rings work in concert to restrict the factors of septum formation at the division site, but the CAR may play a more dominant role in the restriction [51,64]. As described above, although septin rings are not essential for septum formation in fission yeast, they play a crucial role in mediating the timely arrival and departure of glucan synthases, which are required to regulate the proper function of the septum.

In human cells, septins appear to be involved in CAR assembly by bundling and bending actin filaments but septins do not appear to be essential for CAR organization [60,71,72]. However, septins function as a scaffold interacting with myosin IIA and promoting the localization of citron kinase and ROCK2 at the cleavage furrow [71]. Therefore, similar to septins in fission yeast, human septins function to fine-tune cytokinetic events.

## 5. The Roles of Fission Yeast Septins in Septum Degradation

During CAR constriction, glucans populate into the cleavage furrow to form the primary septum, which is flanked by two secondary septa. The functions of the primary and secondary septa are different. The primary septum appears to maintain the stability of the contractile actomyosin ring when the CAR undergoes constriction, while the secondary septa are related to the formation of the cell wall [73,74]. Cell septation begins with the degradation of the primary septum, which is mediated by the α-(1,3) glucanase Agn1 and the β-(1,3) glucanase Eng1 [39,40]. Agn1 and Eng1 are transported to the primary septum by secretory vesicles in an exocyst-dependent manner and form a ring structure flanked by the septin double rings at the septum [44]. Septin rings are required to form Agn1 and Eng1 rings as the absence of Spn1 causes the diffusion of Agn1 and Eng1 within the septum [44]. Additionally, the turnover of Eng1 at the septum depends on Spn1, but the hydrolytic activity of glucanases appears to be normal in cells lacking Spn1 [44]. The fission yeast guanine nucleotide exchange factor (GEF) Gef3 physically interacts with septins and the anillin protein Mid2 and depends on them to localize to the septum where Gef3 may contribute to vesicle tethering and trafficking by interacting with the RhoGTPase Rho3 [55]. It has been reported that Gef3 is a GEF of Rho4, another RhoGTPase, and interacts with septins and activates Rho4 to mediate the proper secretion of the glucanases Eng1 and Agn1 [56,58]. Collectively, it is likely that septins mediate cell separation by serving as positional markers and/or scaffolds to regulate the proper localization of glucanases at the septum (Figure 3B).

Cell septation is executed by glucanases Eng1 and Egt2 and endochitinase Cts1 in budding yeast, and the expression of Eng1 and Cts1 is regulated by the transcriptional factor Ace2 while the expression of Egt2 is regulated by both Ace2 and Swi5 [40,75,76]. However, the role of septins in regulating cell septation in budding yeast is unclear. In mammalian cells, septins play an important role in regulating the last step of cytokinesis (i.e., abscission). As mentioned above, septins are required for intercellular bridge maturation and ESCRT III recruitment to the abscission site [77,78]. However, the exact mechanism of septins in regulating this process is not fully understood [77].

## 6. The Proteins Regulating Septin Assembly during Cytokinesis

In different organisms, the timely assembly and proper organization of septin rings at the cell division site are necessary for cytokinesis. Anillin family proteins play crucial roles in mediating septin organization and function.

Anillin was identified in *Drosophila melanogaster* in 1989 as an actin-binding protein [79,80]. It is a conserved multi-domain protein and appears to link the plasma membrane, septin and microtubule cytoskeletons, and the CAR machinery [79,80]. The N-terminus of anillin contains sites responsible for binding F-actin, myosin, formins, and Rho-dependent kinases, all of which are required to promote cytokinesis. The C-terminus of anillin contains three domains: a Rho-binding domain (RBD), a cryptic domain (C2), and a pleckstrin homology domain (PH), and with these functional domains at the C-terminus, anillin associates with PI(4,5)P_2_ and interacts with RhoA and septins. In mammalian cells, anillin recruits septins to the cleavage furrow to stabilize the CAR and to promote furrow ingression during cytokinesis [60,62,81].

At the onset of cytokinesis in mammalian and budding yeast cells, septins undergo an hourglass-to-double rings (HDR) transition and function as a diffusion barrier to prevent the premature diffusion of cytokinetic components [82,83]. However, how HDR transition occurs has not been fully understood. In budding yeast, anillin-like protein Bud4 and RhoGEF (a guanine nucleotide exchange factor) Bud3 are required to mediate HDR transition. Both Bud4 and Bud3 localize to the outer zones of the septin hourglass before cytokinesis and then to the septin double ring during and after cytokinesis [82]. Specifically, Bud3 functions to promote the assembly and/or maintenance of the circumferential single filaments in the transitional hourglass, which depends on the interaction of Bud3 with Bud4 and septins but does not depend on the GEF activity of Bud3. Bud4 functions to stabilize the transitional hourglass and double septin rings, particularly at the mother side of the bud neck [82].

Similarly, the fission yeast *Schizosaccharomyces pombe* has an anillin-like protein, namely Mid2. Mid2 is a master regulatory protein of septin ring assembly [45,49,50]. Mid2 colocalizes and interacts with septin at the equatorial cortex and forms double rings at the septum during cytokinesis [45,49] (Figure 3B). In cells lacking Mid2, the fluorescence intensity of septins at the septum significantly decreases, and septins fail to compact into a ring structure or split into double rings [49]. Microscopic analysis revealed that the orientation of septin filaments is consecutively disordered in *mid*2Δ cells [49,50]. Furthermore, FRAP results demonstrate that the septin ring becomes more dynamic in *mid*2Δ cells [45]. Therefore, the anillin-like protein Mid2 functions to promote the compaction and stabilization of septin ring structures during cytokinesis.

In addition to anillin family proteins, transcription appears to be involved in regulating septins. The transcriptional factor Ace2 indirectly regulates septins by controlling the expression of *mid*2 [54]. In cells lacking Ace2, Mid2 is insufficient to promote the compaction of the septin ring [54]. Additionally, Spt20, which is a structural subunit of the SAGA (Spt–Ada–Gcn5–acetyltransferase) transcriptional activation complex, was reported to regulate septins in a transcription-dependent and transcription-independent manner [84]. Specifically, Spt20 is required to transcriptionally activate *mid*2 expression. Moreover, Spt20 colocalizes and interacts with Spn2 and Mid2 and is required to stabilize the septin ring and to recruit Mid2 [84]. Hence, septin ring assembly is spatiotemporally regulated by both an anillin family protein and a transcription-dependent manner in fission yeast.

## 7. Septin Localization Is Regulated by Posttranslational Modifications

During the cell cycle, septin localization is regulated by posttranslational modifications. In budding yeast, the p21-activated kinase Cla4 phosphorylates septin Cdc10 upon bud emergence, and the phosphorylation functions to promote the transition of septins from a ring to a collar structure [85]. Gin4, a kinase localizing to the bud neck, phosphorylates septin Shs1 and colocalizes with septins at the bud neck during the cell cycle [86]. When the septin collar splits into double rings, Gin4 dissociates from the septin rings [86]. Cdr2 is the counterpart of Gin4 in fission yeasts but the cortical localization of Cdr2 is independent of septins [57]. In addition to Cla4 and Gin4, several other kinases, including Kcc4 [87], Hsl1 [88], and Elm1 [89,90], have been reported to interact with septins and regulate the structural transition and organization of septins. Phosphorylation is the most frequently reported modification of mammalian septins, which is involved in regulating cytokinesis, sperm annulus organization, neuronal morphogenesis, and post-synaptic stability [91]. Whether and how fission yeast septins are regulated by phosphorylation await further investigation. 

Ubiquitin and small ubiquitin-like modifiers, including SUMO, are also involved in regulating septins. For example, in mammalian cells, RNF8 mediates the ubiquitination of SEPT7 during mitosis and cytokinetic abscission [92]. Additionally, septin mutants that are incapable of modification by SUMO cause the formation of aberrant septin bundles and cytokinesis defects [93]. In budding yeast, SUMO Smt3 associates with septins Cdc3, Cdc11, and Shs1 localizing at the mother site of the bud neck to maintain septin polymerization [94,95].

In summary, posttranslational modifications play crucial roles in regulating septin ring assembly and cytokinesis. However, how posttranslational modifications affect septin assembly and functions and how septins are regulated by posttranslational modifications in space and time await further investigation.

## 8. Perspectives

Septins are conserved GTP-binding proteins. GTP binding, but not GTP hydrolysis, appears to play a role in regulating the self-interaction of septins and the formation of septin filaments [96]. However, the presence of GDP still induces the formation of septin filaments for mammalian SEPT2 [97], suggesting that GTP binding is not necessary for septin filament formation. Therefore, it remains interesting to determine the contributions of the GTPase activity of septins to the organization and assembly of septins.

Another characteristic feature of septins is the capability of membrane binding. Previous studies demonstrated that the polybasic region equips septins with the capability of membrane binding. Our work further demonstrated that Rga6, a Cdc42 GAP protein localizing on the plasma membrane, facilitates the membrane-binding capability of septins [28]. Similarly, in budding yeast, the Cdc42 GAPs Bem3, Rga1, and Rga2 are also involved in the recruitment of septins and the stability of septin ring structures [98]. Interestingly, mammalian ARHGAP4 forms a complex with mammalian septins SEPT2 and SEPT9 via its GAP and SH3 domains while SEPT2 and SEPT9 negatively regulate ARHGAP4 to promote the formation of focal adhesions [99]. Therefore, GAP family proteins seem to play an important role in regulating many aspects of septins, and future studies may focus on addressing the interplay between GAP family proteins and septins.

Septins interact with organelles and regulate organelle functions. In budding yeast, septins regulate ER composition at the bud neck [100]. Specifically, Scs2, a protein localizing to the ER, interacts with septin Shs1, by which an ER diffusion barrier at the bud neck is created and functions to module ER composition [100]. In mammalian cells, septin SEPT2 interacts with Drp1 to promote Drp1-dependent mitochondria fission [101]. Septins can form a cage-like structure to entrap bacteria like *Shigella flexneri* and restrict their dissemination by promoting autophagy [102]. During this process, mitochondria are required to promote the assembly of the septin cage [102]. In fission yeast, glucose starvation not only causes mitochondria fission but also triggers the assembly of filamentous septins in cells lacking septin Spn2 [53]. Whether fission yeast septins are involved in regulating organelles is an interesting question to be explored.

Fission yeast is an excellent model organism to study the cell cycle, cell polarity, cytoskeleton, and organelles, which may involve septins. Indeed, septins are involved in regulating cell polarity, the assembly, maintenance, and constriction of the CAR, and septum degradation in fission yeast. Additionally, they function as diffusion barriers on the cortical membrane to ensure the timely occurrence of cytokinetic events. Although much has been made to understand the functions of fission yeast septins, it remains elusive how septin filaments and rings form on the cortical membrane and how septin functions are orchestrated in space and time. In general, the transition between septin structures during the cell cycle, the structural organization of the purified septin complex, the forces mediating septin motility on the plasma membrane, and the posttranslational modifications of septins are interesting open questions to be explored. In the future, a combination of structural, biochemical, live-cell microscopy, and computational approaches may be useful in tackling the remaining challenges of septin biology.

## Figures and Tables

**Figure 1 jof-10-00115-f001:**
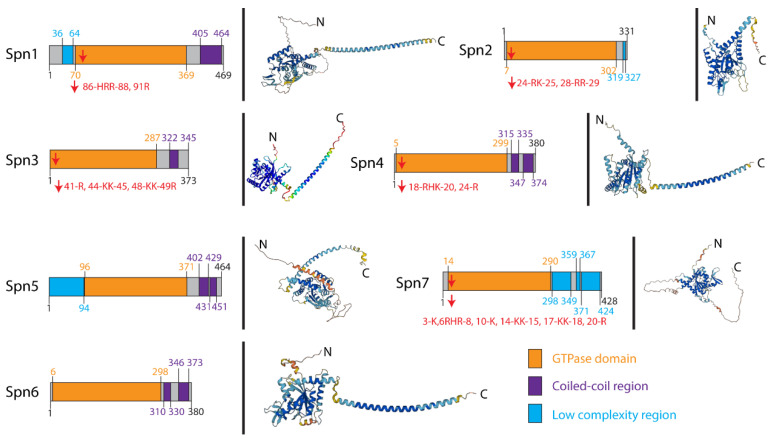
Domains and predicted structures of fission yeast septins. The fission yeast *Schizosaccharomyces pombe* has seven septins, namely Spn1–7. Spn1–4 function in vegetative cells, while Spn2, 5, 6, and 7 form a complex during meiosis to promote the formation of the forespore membrane. The domain structures were drawn according to the annotations in Pombase (www.pombase.org, accessed on 27 November 2023), and structures were predicted by AlphaFold (alphafold.ebi.ac.uk, accessed on 19 November 2023). The indicated domains are colored, and the N-terminal polybasic regions are indicated by arrows (basic residues are shown).

**Figure 2 jof-10-00115-f002:**
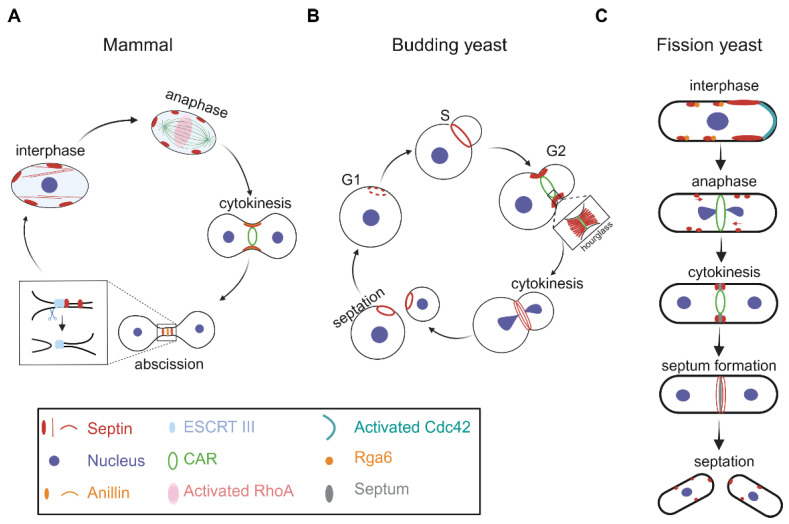
Schematic diagrams illustrating septin localization during the cell cycle. The localization of septins during the cell cycle in mammals (**A**), the budding yeast *Saccharomyces cerevisiae* (**B**), and the fission yeast *Schizosaccharomyces pombe* (**C**). Note that not all fission yeast septins localize to the lateral cortex during interphase. Graphs were generated with BioRender.com (accessed on 15 December 2023).

**Figure 3 jof-10-00115-f003:**
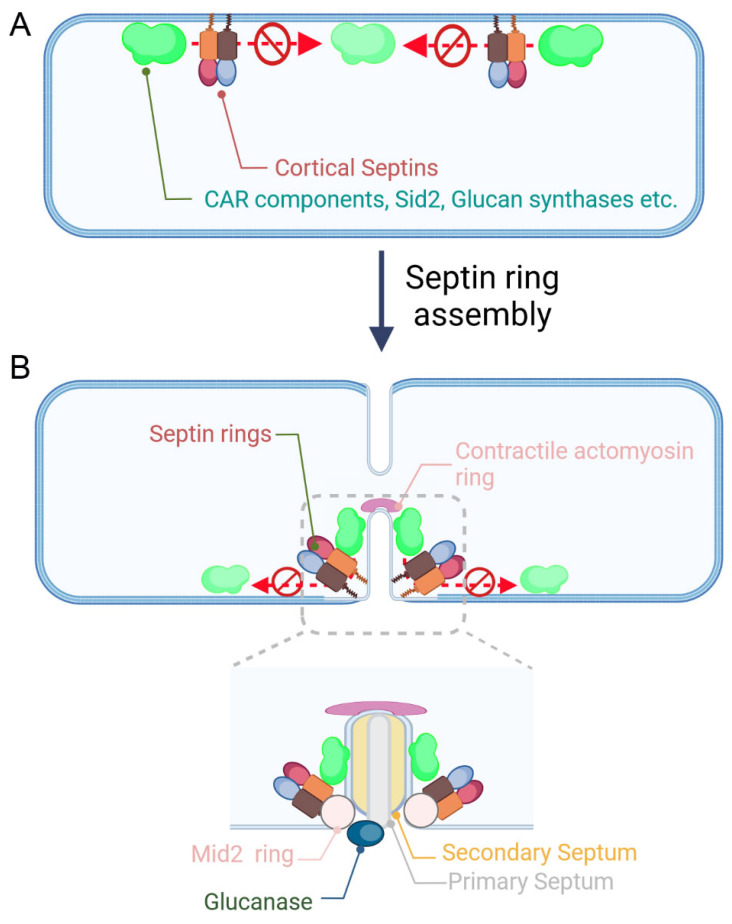
The roles of septins in regulating cytokinesis in fission yeast. (**A**) Before septin ring assembly, septins form puncta structures on the cell cortex and function as barriers to prevent premature accumulation of cortical proteins, including contractile actomyosin ring (CAR) components, Sid2, and glucan synthases, to the equatorial region. (**B**) When the CAR constricts, septin rings are assembled at the equatorial region, which is coordinated by the anillin-like protein Mid2. Septin double rings confine Sid2 and glucan synthases to the cleavage furrow to promote CAR constriction. In addition, septin double rings promote the proper localization of glucanases to the septum, where the primary septum is digested. Graphs were generated with BioRender.com (accessed on 15 December 2023).

## Data Availability

All data are contained within the manuscript.

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
