# Peer review of "The Roles of Septins in Regulating Fission Yeast Cytokinesis"

_jof, 2024, doi:10.3390/jof10020115_

Round 1

Reviewer 1 Report

Comments and Suggestions for Authors

In this manuscript, the authors describe and review the structure and function of septin molecules in the fission yeast Schizosaccharomyces pombe. In fission yeast, septins interact with Cdc42 GAP and aniline and are involved in the localization of glucan synthases. As a result, septins play essential functions in the formation and contraction of CARs and in the regulation of cell polarity. The authors describe the findings on septin in fission yeast in comparison with those in budding yeast and animal cells.

The paper covers a wide range of studies on fission yeast septin and deserves to be published. On the other hand, the following points should be revised.

Major point

I am not a septin expert, but from what I have read, it seems that there is still less knowledge about septins in fission yeast than in budding yeast or animal cells. The authors should clearly state the advantages and significance of studying septins in fission yeast in the introduction and prospect sections. This would make the significance of publishing this paper more profound.

Minor points

1) NC interface (L33) and GG interface (L35) are not clear to readers who are not familiar with Septin. Please add explanations.

2) L35, "self-interact with each other" is difficult to understand what it means. Please rephrase.

3) It is difficult to understand what kind of structure the hourglass-like structure is. It would help the reader if you could show it in Figure 2.

4) For LL 251-253, "At the onset of cytokinesis in mammalian and budding yeast cells, septins undergo an hourglass-to-double rings (HDR) transition and function as a diffusion barrier to prevent premature diffusion of cytokinetic components.please cite references.", please cite references.

Author Response

Point-by-point responses to reviewer 1’s comments

Comments and Suggestions for Authors

In this manuscript, the authors describe and review the structure and function of septin molecules in the fission yeast Schizosaccharomyces pombe. In fission yeast, septins interact with Cdc42 GAP and aniline and are involved in the localization of glucan synthases. As a result, septins play essential functions in the formation and contraction of CARs and in the regulation of cell polarity. The authors describe the findings on septin in fission yeast in comparison with those in budding yeast and animal cells.

The paper covers a wide range of studies on fission yeast septin and deserves to be published. On the other hand, the following points should be revised.

We would like to thank the reviewer for the positive comments.  

Major point

I am not a septin expert, but from what I have read, it seems that there is still less knowledge about septins in fission yeast than in budding yeast or animal cells. The authors should clearly state the advantages and significance of studying septins in fission yeast in the introduction and prospect sections. This would make the significance of publishing this paper more profound.

Yes, the knowledge about septins in fission yeast is limited. We have followed the suggestions to add relevant statements, as below, in the introduction and prospect sections.

In the introduction:

“Fission yeast is a rod-shaped unicellular organism (Hayles and Nurse, 2018), and the rod-shaped morphology makes fission yeast possess two different membrane curvatures: 1) the lateral low-curvature cortex, and the high curvatures at the growing tips and the cleavage furrow. These morphological features make it convenient to study septin behaviors on membranes with different curvatures in one organism. In addition, fission yeast grows by tip elongation (Hayles and Nurse, 2018), and the length of the cell reflects the phase of the cell cycle. Thus, fission yeast is an excellent model to study septins during the cell cycle. Moreover, the highly polarized growth pattern of fission yeast makes it useful to study the roles of septins in regulating cell polarity. Hence, fission yeast is an excellent model organism, complementing budding yeast for studying septin biology.”

In the prospect:

“Fission yeast is an excellent model organism to study cell cycle, cell polarity, cytoskeleton, and organelles, which may involve septins. Indeed, septins are involved in regulating cell polarity, the assembly, maintenance, and constriction of CAR, and septum degradation in fission yeast”

Minor points

1) NC interface (L33) and GG interface (L35) are not clear to readers who are not familiar with Septin. Please add explanations.

We have rephased the indicated as the statements below:

“In addition, septins interact with one another by two different interfaces. 1) The N- and C-termini of two neighboring septins interact with each other to form an NC interface [10, 11]; 2) the two GTP-binding domains of two neighboring septins interact with one another to form a GG interface.”

2) L35, "self-interact with each other" is difficult to understand what it means. Please rephrase.

The indicated sentence has been rephased.

“2) the two GTP-binding domains of two neighboring septins interact with one another to form a GG interface.”

3) It is difficult to understand what kind of structure the hourglass-like structure is. It would help the reader if you could show it in Figure 2.

The hourglass-like structure of septins has been added in revised Figure 2.

4) For LL 251-253, "At the onset of cytokinesis in mammalian and budding yeast cells, septins undergo an hourglass-to-double rings (HDR) transition and function as a diffusion barrier to prevent premature diffusion of cytokinetic components.please cite references.", please cite references.

Two references below have been cited.

(Chen et al., 2020; Ong et al., 2014).

Reviewer 2 Report

Comments and Suggestions for Authors

 In their review, Zheng and colleagues provide a comprehensive summary of the current understanding of septins in fission yeast. The authors repeatedly emphasize that several aspects of septins have not been extensively studied in fission yeast, highlighting the need for further research in this area.  The limited knowledge on the subject, raises the question of the interest of this review for the fission yeast community.

However, to enhance the manuscript, I recommend that the authors incorporate comparisons with budding yeast and other model organisms in the sections devoted to cytokinesis.

Author Response

Point-by-point responses to reviewer 2’s comments

Comments and Suggestions for Authors

In their review, Zheng and colleagues provide a comprehensive summary of the current understanding of septins in fission yeast. The authors repeatedly emphasize that several aspects of septins have not been extensively studied in fission yeast, highlighting the need for further research in this area.  The limited knowledge on the subject, raises the question of the interest of this review for the fission yeast community.

However, to enhance the manuscript, I recommend that the authors incorporate comparisons with budding yeast and other model organisms in the sections devoted to cytokinesis.

Thank you for the suggestion. We have followed the suggestions to add statements of comparisons in the cytokinesis section (Lines 261-290 and 321-330).

Round 2

Reviewer 1 Report

Comments and Suggestions for Authors

I would like to thank the authors for responding to all my comments. I believe that this revision makes the significance of the study of fission yeast in septin biology even clearer.

Author Response

We would llike to thank the reviewer for the support.

Reviewer 2 Report

Comments and Suggestions for Authors

The authors have satisfactorily addressed most of my concerns. The revised manuscript is ready for publication.

Author Response

We would like to thank the reviewer for the kind support.